# Risk Factors for Locomotive Syndrome in Brazilian Older Adults: A Nested Case–Control Study

**DOI:** 10.3390/ijerph22081276

**Published:** 2025-08-15

**Authors:** Julia de Carvalho Galiano, Patricia de Castro Rodrigues, Fania Cristina dos Santos, Virginia Fernandes Moça Trevisani

**Affiliations:** 1Discipline of Geriatrics and Gerontology, Federal University of São Paulo (UNIFESP), São Paulo 04025-002, SP, Brazil; 2Discipline of Emergency Medicine and Evidence-Based Medicine, Postgraduate Program in Evidence-Based Health, Federal University of São Paulo (UNIFESP), São Paulo 04024-002, SP, Brazil

**Keywords:** locomotive syndrome, aged, Brazilian older adults, geriatric locomotive function Scale-25, physical performance tests

## Abstract

This study aimed to describe the risk factors for locomotive syndrome (LS) in Brazilian oldest old individuals. Study subjects were older adults aged ≥80 years who were cognitively healthy, independent for activities of daily living and had been followed since 2011 by a Brazilian cohort study entitled the Longevos Project. A nested case–control study evaluating demographic and clinical characteristics was conducted. Physical tests including the 5 times sit-to-stand, hand-grip strength, 4-m gait speed and two-step test were performed. The World Health Organization Quality of Life Questionnaire short form (WHOQOL-BREF) and Numerical Rating Scale plus Verbal Rating Scale were applied to assess quality of life (QOL) and chronic pain (CP), respectively. LS was diagnosed using the Brazilian version of the 25-question Geriatric Locomotive Function Scale (GLFS-25-p), and sarcopenia by the SARC-F screening tool. The study sample included 52 participants, had a mean age of 89.3 years (±3.9 years) and was predominantly female (76.9%). Of this sample, 24 were diagnosed with LS and 28 were not. The prevalence of osteoporosis (20%), osteoarthritis (22%), depression (13%) and use of a walking device (14%) was higher in the LS group. Participants with LS had worse performance in physical tests. A multivariate logistic regression analysis identified the main risk factors for LS as osteoporosis (OR 10.80, 95%CI 1.08–108.48) and presence of moderate-to-severe chronic musculoskeletal pain (OR 8.92, 95%CI 1.25–63.89).

## 1. Introduction

Living a long life is a triumph. In Brazil, official 2022 data estimate there were more than 32 million people aged 60 years or older, and over 2 million beyond the eighth decade of life [1]. The World Health Organization (WHO) defines healthy aging as the process of development and maintenance of functional capacity that allows well-being in old age. In a dynamic process, the capabilities of older individuals are facilitated or impaired by the environment. The set of attributes of the physical, mental and sensory aspects of older adults sustain intrinsic capacityand are inherent to the individual and their particular characteristics during the aging process. Physical attributes support locomotion and allow the basic human need of moving around and enjoying an independent life. Mobility is associated with the perception of quality of life (QOL) and well-being in old age [2,3,4].

The increase in the aging population is a worldwide phenomenon, where each society has created strategies to deal with this shift [5,6]. In Japan, more than 30% of the population is aged 60 years or older. Since the year 2000, the government of Japan has been developing policies to address the locomotion issue in this population. These public measures have come in response to the high demand for long-term care among older adults with orthopedic problems. In 2007, the Japanese Orthopaedic Association (JOA) defined locomotive syndrome (LS) as the decline in locomotion due to impairment of structures such as muscles and nerves, joints, the spine and intervertebral disks [7,8,9]. LS carries a high risk of walking disability. The main conditions associated with the syndrome are osteoporosis and fragility fractures, osteoarthritis, spondylosis, sarcopenia and neural disorders [10,11,12,13].

Questionnaires and physical function tests are proposed by the JOA for recognizing patients with LS [14]. In 2016, the diagnostic tool called the 25-question Geriatric Locomotive Function Scale (GLFS-25-p) was translated into Brazilian Portuguese [10,15]. In Brazil, the estimated prevalence of LS among octogenarians is 55% [16]. According to social security data, osteoarthritis is the fourth-leading cause of retirement. Osteoporosis affects 20 million people in Brazil and is associated with around 121,700 hip fractures a year. Vertebral fractures were observed in 29% of individuals aged >65 years from São Paulo city [17,18,19,20,21]. All of these pathologies can lead to LS and dependence on care. When an older individual loses functional independence, they are more susceptible to becoming confined to the home or institutionalized. There are an estimated 45,000 older residents in long-term care facilities in Brazil. Several reasons prompt the institutionalization of this population, one of which is functional dependency [8,11,21,22].

Scientifically, it is important to raise awareness of LS in Brazil to facilitate health policies and help prevent functional decline in aging people or the need for long-term care. This study aimed to identify the risk factors associated with LS in Brazilian elderly people.

## 2. Materials and Methods

A nested case–control study of outpatients was carried out at the Federal University of São Paulo, a public health service in São Paulo city, Brazil. Participants had previously taken part in a follow-up cohort study entitled the Longevos Project that started in 2011 [23]. The present investigation was conducted from April to November 2021. The study design was viable because it was quick and practical, as it was carried out in a center with a large flow of elderly people, especially the oldest old.

The LS cases were diagnosed using the GLFS 25-p, adopting a cut-off score of ≥19 points [16]. Control subjects were selected from the group scoring <19 points on the scale from the same site and were matched for characteristics, such as age, and drawn from the Longevos Project population. The Brazilian GLFS-25 cut-off score of 19 was defined by comparing it with other functional mobility tests using the mosaic plot described in the Arbex et al. study. Subsequent ROC analysis for the GLFS-25p score 19 obtained a sensitivity of 0.86, a specificity of 0.67, and an area under the ROC curve of 0.83 [15].

The local Research Ethics Committee approved the study. Inclusion criteria comprised being an older adult (aged ≥ 80 years) of either gender, being independent for activities of daily living and for walking (irrespective of the use of an assistive device such as a walker or cane). Exclusion criteria comprised hospitalization in the last year (up to one year before data collection) or diagnosis with dementia. Individuals with uncontrolled diseases or undergoing parenteral chemotherapy or radiotherapy were also excluded.

The data were collected using a semi-structured protocol. The information collected included sociodemographic data (age, gender), personal history, medications in use, occurrence of previous fragility fracture, falls in last year, and use of assistive walking devices. Data on anthropometry, data on physical activity (≥150 min/week) and pain assessment were collected. Scores on the GLFS 25-p and QOL scale (World Health Organization Quality of Life Questionnaire short form—WHOQOL-BREF) were also collected [24]. Physical performance was assessed using the 4-m gait speed [25], 5 times sit-to-stand test [26], two-step test (TSS) [8], and hand-grip strength test [27], and by sarcopenia screening with the Brazilian version of the SARC-F tool [28]. Relevant information was also taken from medical records.

### 2.1. Measurements

#### 2.1.1. Definition of Locomotive Syndrome

The Brazilian validated version of the GLFS 25-p consists of 25 questions divided into domains that comprise questions relating to pain, activities of daily living, social performance and mental health status. Each question is graded on 5-point scale from 0 to 4 points. The sum of total scores ranges from 0 to 100. The cut-off score for identifying LS cases in Brazil is 19 points, as established by Arbex et al. [14,16].

#### 2.1.2. Quality of Life and Chronic Pain

QOL assessment was performed using the WHOQOL-BREF. The short form questionnaire covers four domains of QOL: physical health (7 items), psychological health (6 items), social relationships (3 items) and environmental health (8 items). The scale also covers 2 other items on overall QOL and general health items, giving a total of 24 facets and 26 questions. All of these items are rated on a 5-point scale, with higher scores indicating better QOL. Final scores on each domain are calculated by a syntax that compares linearly against a scale of 0 to 100 [24].

Chronic pain (CP) (for a period of at least 3 months) was classified based on location and intensity. Intensity was rated by the Numerical Rating Scale (NRS) and Verbal Rating Scale (VRS) [29]. For the NRS, patients were asked to rate their pain between 0 and 10, with 0 representing “no pain” and 10 indicating “the worst pain imaginable”. For the VRS, patients rated intensity as 0 for “no pain”, 1 for “mild pain”, 2 for “moderate pain” and 3 for “severe pain”. The Portuguese version of the Douleur Neuropathique 4 (DN4) questionnaire was applied to identify cases of neuropathic pain. The cut-off score of 4 points suggests neuropathic pain [30]. Cases of musculoskeletal pain were analyzed by body segment: upper limb (arms, shoulders and neck); lower limb (hips, knees and legs); and spine (thoracic or lumbar). Pain of non-musculoskeletal origin was not analyzed. Intensity of pain was divided into two categories: no/mild pain and moderate-to-severe pain on the VRS, while pain on the NRS was assessed as a continuous variable.

#### 2.1.3. Physical Performance and Sarcopenia Screening

The 4 m-gait Speed test: participants were asked to walk at their normal pace along a linear path of 4 m in a flat area twice. The fastest time taken out of the 2 tries was selected. A cut-off speed of 0.8 m/s was defined, where values below this cut-off denote poor performance [25].

The 5 times sit-to-stand test: keeping both arms folded across their chest, participants were asked to stand and sit 5 times while being timed with a stopwatch. For participants aged ≥80 years, the cut-off time was 14.8 s. If the individual was unable to get up from the chair 5 times, no score was assigned [26].

The two-step test (TSS): The participant was instructed to take two strides forward to the maximum extent possible without losing balance. Use of their usual walking assistive device was allowed. The maximum length of the 2 strides (cm) was divided by the individual’s height (cm). No cut-off point has been defined for Brazilian older adults, but values < 1.1 for Japanese adults indicate impairment of lower limb mobility [8].

The hand-grip strength (HG) test: a Jamar^®^ hydraulic dynamometer (Sammons Preston, Bolingbrook, IL, USA) was used, together with the application protocol defined by the American Society of Hand Therapy. The dominant hand was tested with the participant sitting in the upright position, and the highest value out of 3 tries was used [27].

The SARC-F test: participants answered a 5 question tool that measures the ability to carry weight, walk unaided, transfer from chair or bed, climb stairs, and number of falls. Each item was scored from 0 to 2, with 0 for no difficulty, 1 for some difficulty and 2 for failure/unable. A score ≥6 is suggestive of sarcopenia [28].

### 2.2. Statistical Analysis

Data were expressed as absolute and relative frequencies and means of summary measures. Associations were analyzed using the Chi-square test or, in cases of small samples, Fisher’s Exact test. Adjusted standardized residuals were used to identify local differences, where absolute values >1.96 indicated evidence of associations between categories. Comparisons of means was performed using Student’s *t*-test, followed by verification using the Kolmogorov–Smirnov test or non-parametric Mann–Whitney test.

The effects of demographic and clinical characteristics on the occurrence of LS were assessed by fitting univariate and multivariate logistic regressions. Due to the number of predictive variables compared to the sample size, variables whose associations with the dependent variable were significant at 10% on the univariate analysis were selected for entry into the initial multivariate model. Subsequently, variables not significant at 5% were excluded stepwise in order of significance (backward method). In addition, the goodness-of-fit of the final model was evaluated using the Hosmer–Lemeshow test. A significance level of 5% was adopted for all statistical tests. Statistical analyses were carried out using the SPSS 20.0 (IBM SPSS Statistics Base, Armonk, NY, USA) and STATA 17 (Stata Statistical Software: Release 17, College Station, TX, USA) statistical software packages.

## 3. Results

Of the 52 participants, 24 were diagnosed with LS and 28 were not. The overall sample had a mean age of 89.3 (SD ± 3.9) years and was predominantly female (76.9%). LS prevalence was higher in women than men (91.7% vs. 8.3%). Demographic and clinical data are shown in Table 1. The rates of osteoarthritis, osteoporosis, depression and need for a walking device were higher in the LS group. No association with the other characteristics was found.

Subjects without LS attained better results on the 4-m gait speed, 5 times sit-to-stand and TSS tests (Table 2). Adopting the Japanese cut-off, performance on the TSS was not associated with LS. Sarcopenia and CP rates were lower in the group without LS.

Regarding CP, patients reported CP affecting the neck, shoulders, lumbar spine, hips, knees, legs and abdomen. A positive association was found between LS and CP intensity, as classified by the NRS and VRS. There was one case of neuropathic pain in the thoracic region and one case of chronic headache. WHOQOL-BREF—physical domain scores were worse in the LS group, while no difference was evident for the other domains (Table 3).

Logistic regression results are presented in Table 4 and Table 5. Osteoporosis (*p* = 0.043), WHOQOL—physical health (*p* = 0.003), moderate or severe CP (*p* = 0.029) and TSS (*p* = 0.016) performance remained significant in the final model. Thus, the risk of LS in patients with osteoporosis was 10.8 times greater than in patients without this condition. Also, the risk of LS in patients with moderate or severe CP was 8.9 times greater than in subjects with mild pain or no pain. In addition, for every 1 point increase on WHOQOL-BREF—physical health, the odds of LS were lower (10% (1.00–0.90) × 100%). Similarly, for every 1 unit increase on the TSS, the odds of LS were lower (99.9% (1.000–0.0007) × 100%). According to the Hosmer-Lemeshow test, the model exhibited a good fit (*p* = 0.141).

## 4. Discussion

The present study comprised oldest old participants without cognitive dysfunction who were independent for activities of daily living. Results showed higher LS prevalence in women than men (*p* = 0.019). Similarly, data from a Japanese population found that, among individuals aged 80 years or older, 76% of women had LS versus 62% of men. Furthermore, people from urban regions had an 80% higher risk of LS, which was even greater among women [31,32]. Motor function declines with age, and recognizing LS is important because this condition results in reduced mobility, greater dependence and higher demand for long-term care [7]. The Pan American Health Organization predicts that long-term care demands may triple in the next three decades [2]. Meanwhile, the number of older Brazilians needing long-term care (at home or in an institution) has increased, yet support provided by public health services remains limited [22,33].

Three tools for identifying LS were proposed by the JOA, but some diagnostic cut-off limits adopted in Japan differ from those used in Brazil [9,10,16]. The GLFS-25 assesses symptoms and locomotor function and adopts 16 points as the cut-off for the Japanese population. The TSS identifies lower limb function in a horizontal trajectory, while the stand-up test assesses muscle power and locomotor function for a vertical task. On the JOA screening test, the ability to stand with a double-leg or single-leg stance from stools of different heights is evaluated [8]. Tasks involving sitting and standing rely on strength of the lower limbs and aerobic capacity of the subject and, for the 5 times sit-to-stand test, meta-analysis studies have validated the cut-off according to age [25,26,34]. In a Japanese study, more than 96% of the participants aged over 80 years were unable to perform the JOA stand-up test, although individuals aged ≥90 years were not included [35]. Compared with the present study sample, 86% of participants were able to perform the 5 times sit-to-stand test, even those with LS. Arbex et al., based on the results of the LOCOMOV Project, found that performance levels on the 4-m gait speed and the 5 times sit-to-stand tests were strongly correlated with each other and with the TSS [16,36]. The TSS cut-off in older Brazilians might also differ because of differences in body composition and functional capacity between Japanese and Brazilian older adults [37].

An article from the University of Tokyo reported that fragility fractures and falls rank fourth among the conditions that most cause disability and lead to the need for support for activities of daily living [31]. By contrast, in the current study, the presence of falls and personal history of fractures were not significant for LS, but osteoporosis was found to be a risk factor for the syndrome (OR, 10.80; 95%CI, 1.08–108.48). According to the epidemiologic study on LS by Yoshimura et al., the prevalence of osteoporosis in participants aged over 80 years was 50% [31]. New studies are needed to establish the causality between osteoporosis and LS.

LS occurs due to impairment to structures of the locomotor organs. In the absence of significant deleterious effects, the affected individual is often unaware of decline in their locomotion. These degenerative processes can accompany the individual for years, progressively worsening their locomotor ability by causing pain, deformities and loss of muscle strength [7]. In the current sample, the presence of CP and its location were associated with LS. Greater musculoskeletal pain intensities were also associated with LS (OR 8.92, 95%CI 1.25–63.89). Iizuka et al. demonstrated that musculoskeletal pain located in the lower lumbar region (OR 2.60, 95%CI 1.29–5.24), knees (OR 2.97, 95%CI 1.41–6.28) and shoulders (OR 2.16, 95%CI 1.00–4.66) was also significant for LS [38].

Pain prevalence increases with age and reduces QOL [39]. Appropriate interventions can be proposed for treatment and rehabilitation, considering the hopes, expectations and feelings of each individual patient as the main outcome [40,41]. There is also a strong association between QOL and social participation [42]. In 2020, the COVID-19 pandemic worsened social life of the older population, while lockdowns during the pandemic negatively impacted activities of daily living. A review study showed that older individuals experienced social isolation and reduced QOL [43]. Another recent publication showed that the development of LS may have increased 2.4 fold in people aged 75 years or older during the pandemic period, especially among sedentary individuals [44]. The results of the present study underscore the importance of QOL measurement, since older participants with higher physical health scores on the WHOQOL-BREF proved to be less affected by LS (OR, 0.90; 95%CI, 0.84–0.96). The present study highlights the importance of other factors in addition to physical tests. The additional factors evaluated in this study, quality of life and CP, demonstrated a greater association with LS. An Indian study demonstrated the relationship between advancing age, reduced quality of life and a comparison between individuals with and without CP. In elderly people with pain, the perception of quality of life was worse. New controlled studies are needed to clarify whether the treatment of CP can provide improvements in locomotion and quality of life [39].

The current study’s results contribute to LS research in that the population investigated comprised oldest old individuals. Knowing the risk factors for LS is important for preventing functional decline and delaying the need for care. The fact that older adults all over the world were affected during the pandemic might constitute a limitation of the present study and a cause of bias. Other limitations include the small sample size and observational study design. Future strategies to tackle LS are needed, particularly in Brazil.

## 5. Conclusions

The present study demonstrated that osteoporosis and moderate-to-severe chronic pain were risk factors for LS in Brazilian older adults.

## Figures and Tables

**Table 1 ijerph-22-01276-t001:** Participant characteristics.

	Without LS (*n* = 28)	With LS (*n* = 24)	Total (*n* = 52)	*p*-Value
Gender				0.019
Female	18 (64.3)	22 (91.7)	40 (76.9)	
Male	10 (35.7)	2 (8.3)	12 (23.1)	
Age (years)				0.136 ^b^
Mean ± SD	88.6 ± 3.9	90.2 ± 3.9	89.3 ± 3.9	
Median (Min—Max)	89.0 (83.0–101.0)	90.5 (84.0–98.0)	89.0 (83.0–101.0)	
BMI (kg/m^2^)				0.549 ^b^
Mean ± SD	25.3 ± 3.9	26.0 ± 4.3	25.7 ± 4.0	
Median (Min—Max)	25.0 (18.4–34.7)	24.8 (18.6–35.8)	24.9 (18.4–35.8)	
Hypertension				0.728 ^a^
No	6 (23.1)	4 (16.7)	10 (20.0)	
Yes	20 (76.9)	20 (83.3)	40 (80.0)	
Diabetes mellitus				0.877
No	19 (73.1)	18 (75.0)	37 (74.0)	
Yes	7 (26.9)	6 (25.0)	13 (26.0)	
Previous stroke				0.103 ^a^
No	26 (100.0)	21 (87.5)	47 (94.0)	
Yes	0 (0.0)	3 (12.5)	3 (6.0)	
Chronic renal failure				0.182
CKD-EPI ≥ 30	18 (64.3)	11 (45.8)	29 (55.8)	
CKD-EPI < 30	10 (35.7)	13 (54.2)	23 (44.2)	
Osteoarthritis				0.019
No	10 (35.7)	2 (8.3)	12 (23.1)	
Yes	18 (64.3)	22 (91.7)	40 (76.9)	
Osteoporosis				0.023
No	13 (46.4)	4 (16.7)	17 (32.7)	
Yes	15 (53.6)	20 (83.3)	35 (67.3)	
Previous fracture				0.335
No	23 (82.1)	17 (70.8)	40 (76.9)	
Yes	5 (17.9)	7 (29.2)	12 (23.1)	
Depression				0.031
No	21 (75.0)	11 (45.8)	32 (61.5)	
Yes	7 (25.0)	13 (54.2)	20 (38.5)	
Falls				0.358 ^a^
<2	24 (85.7)	23 (95.8)	47 (90.4)	
≥2	4 (14.3)	1 (4.2)	5 (9.6)	
Polypharmacy				0.054
0 to 4	10 (35.7)	3 (12.5)	13 (25.0)	
≥5	18 (64.3)	21 (87.5)	39 (75.0)	
Physical activity				0.157
No	20 (71.4)	21 (87.5)	41 (78.8)	
Yes	8 (28.6)	3 (12.5)	11 (21.2)	
Walking device				0.006
No	22 (78.6)	10 (41.7)	32 (61.5)	
Yes	6 (21.4)	14 (58.3)	20 (38.5)	

*p*—Chi-square test, Fisher’s exact test (^a^), Student’s *t*-test (^b^). CKD-EPI: equation for estimated glomerular filtration rate.

**Table 2 ijerph-22-01276-t002:** Physical performance and pain evaluation.

	Without LS (*n* = 28)	With LS (*n* = 24)	Total (*n* = 52)	*p*-Value
4-m gait speed				0.002
<0.8 m/s	10 (35.7)	19 (79.2)	29 (55.8)	
≥0.8 m/s	18 (64.3)	5 (20.8)	23 (44.2)	
5 times sit-to-Stand				0.006 ^a^
˂14.8 s	20 (71.4)	10 (41.7)	30 (57.7)	
≥14.8 s	8 (28.6)	7 (29.2)	15 (28.8)	
Unable	0 (0.0)	7 (29.2)	7 (13.5)	
Chronic pain				0.016
No	12 (42.9)	3 (12.5)	15 (28.8)	
Yes	16 (57.1)	21 (87.5)	37 (71.2)	
Musculoskeletal chronic pain				0.012
No	14 (50.0)	4 (16.7)	18 (34.6)	
Yes	14 (50.0)	20 (83.3)	34 (65.4)	
Location of chronic pain				0.021 ^a^
No pain	14 (50.0)	4 (16.7)	18 (34.6)	
Lower limbs	4 (14.3)	12 (50.0)	16 (30.8)	
Spine	6 (21.4)	5 (20.8)	11 (21.2)	
Upper limbs	4 (14.3)	3 (12.5)	7 (13.5)	
VRS				0.008
No pain/Mild pain	16 (57.1)	5 (20.8)	21 (40.4)	
Moderate/Severe	12 (42.9)	19 (79.2)	31 (59.6)	
NRS				<0.001 ^b^
Mean ± SD	3.5 ± 2.6	6.5 ± 2.4	25.7 ± 4.0	
Median (Min—Max)	4.5 (0.0–8.0)	6.5 (0.0–10.0)	24.9 (18.4–35.8)	
TSS				0.005 ^b^
Mean ± SD	0.9 ± 0.20	0.7 ± 0.3	0.8 ± 0.3	
Median (Min—Max)	0.9 (0.6–1.6)	0.8 (0.0–1.1)	0.9 (0.0–1.6)	
TSS Japanese cut-off				0.358 ^a^
<1.1	24 (85.7)	23 (95.8)	47 (90.4)	
≥1.1	4 (14.3)	1 (4.2)	5 (9.6)	
Hand-grip strength				0.278 ^b^
Mean ± SD	23.9 ± 8.0	21.5 ± 7.7	22.8 ± 7.9	
Median (Min—Max)	23.0 (12.0–48.0)	22.0 (2.0–38.0)	22.0 (2.0–48.0)	
SARC-F				0.039 ^a^
Negative	28 (100.0)	20 (83.3)	48 (92.3)	
Positive	0 (0.0)	4 (16.7)	4 (7.7)	

*p*—Chi-square test, Fisher’s exact test (^a^), Student’s *t*-test (^b^). VRS: Verbal Rating Scale. NRS: Numerical Rating Scale. TSS: two-step test. SARC-F: sarcopenia screening tool.

**Table 3 ijerph-22-01276-t003:** Quality of life assessment.

	Without LS (*n* = 28)	With LS (*n* = 24)	Total (*n* = 52)	*p*-Value
WHOQOL—Physical domain				<0.001 ^b^
Mean ± SD	72.6 ± 15.1	50.9 ± 17.9	62.6 ± 19.6	
Median (Min—Max)	71.4 (42.9–100.0)	53.6 (14.3–78.6)	64.3 (14.3–100.0)	
WHOQOL—Psychological health				0.195 ^b^
Mean ± SD	69.8 ± 14.9	64.1 ± 16.6	67.1 ± 15.8	
Median (Min—Max)	75.0 (45.8–95.8)	62.5 (29.2–87.5)	68.8 (29.2–95.8)	
WHOQOL—Social relationships				0.745 ^b^
Mean ± SD	79.2 ± 17.2	77.8 ± 12.7	78.5 ± 15.2	
Median (Min—Max)	83.3 (33.3–100.0)	75.0 (41.7–100.0)	79.2 (33.3–100.0)	
WHOQOL—Environmental health				0.515 ^b^
Mean ± SD	70.8 ± 14.5	68.1 ± 14.7	69.5 ± 14.5	
Median (Min—Max)	75.0 (37.5–90.6)	68.8 (21.9–93.8)	71.9 (21.9–93.8)	
WHOQOL—Overall QOL and general health				0.992 ^c^
Mean ± SD	68.8 ± 16.1	66.7 ± 20.4	67.8 ± 18.1	
Median (Min—Max)	75.0 (50.0–100.0)	75.0 (25.0–100.0)	75.0 (25.0–100.0)	

*p*—Chi-square test, Student’s *t*-test (^b^), Mann-Whitney (^c^). WHOQOL: World Health Organization Quality of Life Questionnaire-short form—WHOQOL-BREF.

**Table 4 ijerph-22-01276-t004:** Results of the univariate logistic regression model.

	OR Crude (95%CI)	*p*-Value
Age (years)	1.12 (0.96–1.30)	0.140
Hypertension	1.50 (0.37–6.14)	0.573
Diabetes mellitus	0.90 (0.25–3.21)	0.877
Previous stroke	(1)	0.999
Chronic renal failure (CKD-EPI < 30)	2.13 (0.70–6.48)	0.184
Osteoarthritis	6.11 (1.18–31.54)	0.031
Osteoporosis	4.33 (1.17–15.99)	0.028
Previous fracture	1.89 (0.51–7.00)	0.338
Depression	3.55 (1.10–11.46)	0.034
Obesity	1.20 (0.27–5.42)	0.813
≥2 falls	0.26 (0.03–2.51)	0.245
Polypharmacy	3.89 (0.93–16.34)	0.064
Physical activity	0.36 (0.08–1.54)	0.167
Gait speed (m/seg)	0.01 (0.00–0.13)	0.001
Gait speed < 0.8 m/s (ref.≥ 0.8 m/s)	6.84 (1.96–23.93)	0.003
5 times sit-to-stand (seg)	0.91 (0.82–1.01)	0.070
5 times sit-to-stand (ref.= 14.8 seg ou menos)		0.687
5 times sit-to-stand > 14.8 seg	1.75 (0.49–6.21)	0.387
5 times sit-to-stand unable	(1)	0.999
WHOQOL—Physical domain	0.92 (0.87–0.97)	0.001
WHOQOL—Psychological health	0.98 (0.94–1.01)	0.193
WHOQOL—Social relationships	0.99 (0.96—1.03)	0.740
WHOQOL—Environmental health	0.99 (0.95–1.03)	0.508
WHOQOL—overall QOL and general health	0.99 (0.96–1.02)	0.676
Chronic pain	5.25 (1.27–21.78)	0.022
Musculoskeletal chronic pain	5.00 (1.36–18.43)	0.016
Location of chronic pain (ref. Sem dor)		0.037
Lower limbs	10.50 (2.15–51.28)	0.004
Spine	2.92 (0.57–14.82)	0.197
Upper limbs	2.62 (0.41–16.93)	0.310
Moderate/Severe NRS	5.07 (1.47–17.46)	0.010
Two step test	0.01 (0.00–0.44)	0.015
Two step test ≥ 1.1 (ref.< 1.1)	0.26 (0.03–2.51)	0.245
Hand-grip strength	0.96 (0.89–1.03)	0.278
SARC-F	(1)	0.999

**Table 5 ijerph-22-01276-t005:** Results of initial and final logistic regression models.

	Initial Model		Final Model	
	Adjusted OR (95%CI)	*p*-Value	Adjusted OR (95%CI)	*p*-Value
Osteoarthritis	1.88 (0.16–22.45)	0.618	-	-
Osteoporosis	9.59 (0.64–143.05)	0.101	10.80 (1.08–108.48)	0.043
Depression	0.97 (0.10–9.06)	0.981	-	-
Polypharmacy	0.85 (0.06–13.09)	0.909	-	-
4-m gait speed	0.05 (0.00–85.73)	0.426	-	-
5 times sit-to-stand	0.94 (0.77–1.14)	0.526	-	-
WHOQOL—Physical Health	0.91 (0.84–0.99)	0.023	0.90 (0.84–0.96)	0.003
Chronic pain	1.23 (0.04–33.85)	0.903	-	-
Moderate/severe pain	6.73 (0.49–93.01)	0.155	8.92 (1.25–63.89)	0.029
TSS	0.01 (0.00–112.48)	0.329	0.0007 (0.000002–0.2612)	0.016

OR: odds ratio. WHOQOL: World Health Organization Quality of Life Questionnaire-short form—WHOQOL-BREF. TSS: two-step test.

## Data Availability

Data presented in this study are available on request from the corresponding author. Data are not publicly available due to privacy.

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
