# Peer review of "Risk Factors for Locomotive Syndrome in Brazilian Older Adults: A Nested Case–Control Study"

_ijerph, 2025, doi:10.3390/ijerph22081276_

Round 1
Reviewer 1 Report
Comments and Suggestions for Authors
I believe this is an important paper that thoroughly investigates Locomotive Syndrome (LS) in the Brazilian population. However, there are some points that require improvement:
Major points
- There is a significant flaw in the multivariate logistic regression analysis. Since the outcome of LS is only 24, the number of explanatory variables should be limited to 2-3. There is a risk of overfitting and multicollinearity. Please also show the results of the univariate analysis. The Initial model seems to be of little value.
Minor points
- Line 74, Line 100: In the text, it states "GLFS 25-p adopting a cut-off score of ≥19," but the Japanese standard is 16 points or higher for LS. Please provide a brief explanation of why 19 points was adopted in this study.
- Line 69-93: Can you categorize this paragraph for better clarity?
- Line 139-142: SARC-F is a self-reported questionnaire, not a physical performance test.
- Line 191: It is more common to use "Odds ratio" instead of expressing Odds as "times."
Author Response
Major points:
Comments 1. There is a significant flaw in the multivariate logistic regression analysis. Since the outcome of LS is only 24, the number of explanatory variables should be limited to 2-3. There is a risk of overfitting and multicollinearity. Please also show the results of the univariate analysis. The Initial model seems to be of little value.
Response 1: Dear reviewer, thank you for your comments regarding the statistical analysis, we agree with your statement. As suggested, we have included the results of the univariate analysis in the paper, you can see them in Table 4 in the revised paper, starting from line 192.
Minor points:
Comments 1. Line 74, Line 100: In the text, it states "GLFS 25-p adopting a cut-off score of ≥19," but the Japanese standard is 16 points or higher for LS. Please provide a brief explanation of why 19 points was adopted in this study.
Response 1: We appreciate and agree with the statement, the explanation regarding the GLFS-25p cutoff point has been included in lines 78-81.
Comments 2: Line 69-93: Can you categorize this paragraph for better clarity?
Response 2: We appreciate and agree with the statement, changes have been made to bring clarity to the text, line 68.
Comments 3: Line 139-142: SARC-F is a self-reported questionnaire, not a physical performance test
Response 3: We appreciate and agree with the statement, changes have been made, which can be seen in line 124.
Comments 4: Line 191: It is more common to use "Odds ratio" instead of expressing Odds as "times."
Response 4: We appreciate and agree with the statement, changes have been made in lines 194-195.
Reviewer 2 Report
Comments and Suggestions for Authors
The study clearly highlights the association of osteoporosis and chronic pain with LS, which is clinically valuable.
Reliable metrics like WHOQOL-BREF and GLFS-25 were used, ensuring the study's precision.
However this study has areas for Improvement:
- Small Sample Size: The study only included 52 participants, which limits the generalizability of the findings. Larger studies are needed for validation.
- Lack of Causality Proof: Being an observational study, it cannot definitively establish causal relationships. Randomized controlled trials would be helpful for further investigation.
No problem.
Author Response
Comments 1: Small Sample Size: The study only included 52 participants, which limits the generalizability of the findings. Larger studies are needed for validation.
Response 1: Dear reviewer, we appreciate your contributions to our article. Unfortunately, the sample size was small due to the particularities of the study population, elderly, functional and with preserved cognition at 80 years of age or older. We agree that larger studies are needed.
Comments 2: Lack of Causality Proof: Being an observational study, it cannot definitively establish causal relationships. Randomized controlled trials would be helpful for further investigation.
Response 2: We agree that studies that demonstrate causality are necessary, and we encourage the development of new research, especially in Brazil and developing countries. We are grateful.
Reviewer 3 Report
Comments and Suggestions for Authors
At a time when aging is rapidly progressing worldwide, I think this is an important study comparing the motor function and physical condition of the elderly.
I have a few questions regarding this.
1. The purpose of this study is unclear. Please clearly state the purpose of this study in the introduction.
2. In a nested case–control study, the statistical method conditional logistic regression is considered very important.
1) Therefore, please briefly mention the reason why you chose the nested case–control study as the research method.
2) I think there is insufficient discussion on Table 4 obtained through conditional logistic regression in the review. Please add additional significance to the items (osteoporosis, WHOQOL, CP, etc.) whose results were significant.
Author Response
At a time when aging is rapidly progressing worldwide, I think this is an important study comparing the motor function and physical condition of the elderly.
I have a few questions regarding this.
Comments 1: The purpose of this study is unclear. Please clearly state the purpose of this study in the introduction.
Response 1: Dear reviewer, thank you for your contribution to our article. We agree with your statement and have adjusted the purpose of the study in the introduction, please see line 66.
In a nested case–control study, the statistical method conditional logistic regression is considered very important.
Comments 2.1) Therefore, please briefly mention the reason why you chose the nested case–control study as the research method.
Response 2.1) We agree with your statement and have adjusted a brief reason for choosing the study in the methodology, please see line 71.
Comments 2.2) I think there is insufficient discussion on Table 4 obtained through conditional logistic regression in the review. Please add additional significance to the items (osteoporosis, WHOQOL, CP, etc.) whose results were significant.
Response 2.2) Dear reviewer, thank you for your contribution to this topic. We agree that these are important aspects to be highlighted. However, the article was clear in its results and it was necessary to be more objective in describing these points due to the character limit.
Round 2
Reviewer 1 Report
Comments and Suggestions for Authors
No further comments.
Author Response
Dear reviewer, thank you for your reviews and contributions.